# Comparison of Efficacy of Intragastric Balloon Devices as Bridging Therapy Prior to Laparoscopic Sleeve Gastrectomy

**DOI:** 10.3390/jcm14217602

**Published:** 2025-10-27

**Authors:** Tomasz Klimczak, Krzysztof Słowiński, Alicja Majos, Jacek Śmigielski, Wojciech Ciesielski

**Affiliations:** 1Department of General and Transplant Surgery, Medical University of Lodz, 90-153 Lodz, Poland; tjjklimczak@gmail.com (T.K.); krzysztofslowinski1993@gmail.com (K.S.); alicja.majos@umed.lodz.pl (A.M.); 2Department of General and Oncological Surgery, Provincial Hospital in Sieradz, 98-200 Sieradz, Poland; smiglo@mp.pl

**Keywords:** intragastric balloon, bridge therapy, super-obesity, laparoscopic sleeve gastrectomy, weight loss outcomes

## Abstract

**Background:** Bridge therapy before surgery is one of indications for intragastric balloon (IGB) implantation. We aim to compare the outcomes of Medsil (non-adjustable), Orbera365 (non-adjustable), and Spatz3 (adjustable) IGBs used as a bridge therapy before laparoscopic sleeve gastrectomy (LSG). **Methods:** The data of 148 patients with super-obesity (BMI > 50 kg/m^2^) who underwent IGB implantation as bridge treatment prior to LSG between July 2018 and December 2022 were analyzed. Patients were allocated according to device availability in consecutive procurement periods: Orbera365 (47 patients), Medsil (53 patients), and Spatz3 (48 patients). Weight loss (kg), BMI reduction, and percentage of excess weight loss (%EWL) were measured at 6 months. **Results:** Weight loss after 6 months was greatest in the Spatz3 group (mean 25 kg; median 24.64 kg; 19.46–33.04 kg) compared to the Medsil (mean 16 kg; median 16 kg; 11.7–33 kg) and Orbera365 (mean 14 kg; median 14.53 kg; 11.54–18.26 kg) groups. %EWL and %TWL were also greatest in the Spatz3 group (%EWL: 22.98%; %TWL: 14.0%) compared to the Medsil (%EWL: 15.06%; %TWL: 9.4%) and Orbera365 (%EWL: 13.71%; %TWL: 8.3%) groups. **Conclusions:** In super-obese patients undergoing a 6-month bridging therapy before LSG, an adjustable IGB with mid-term volume increase achieved greater short-term weight loss than non-adjustable devices. Implications for peri-operative outcomes require confirmation in prospective studies.

## 1. Introduction

Obesity is spreading across both developed and developing countries. It often requires lifestyle changes, including diet, exercise, and psychological support, which may only yield a modest weight loss of 3–4%, necessary for delaying the onset of diabetes and other obesity-related complications [1]. When traditional methods fail, options like the intragastric balloon (IGB) come into play. Introduced in the U.S. in 1984, the IGB is a device placed in the stomach to induce satiety, helping reduce food intake [1,2,3].

Initially, the Garren-Edwards Bubble was a widely used IGB, but due to poor performance and complications, it was discontinued. This led to the 1987 “Obesity and the Gastric Balloon” conference, setting new standards for IGB design and use, including BMI thresholds [3]. Subsequent devices like the saline-filled BIB (Orbera), developed in 1991, saw widespread international use, despite not being approved in North America [2,3].

Over the years, multiple countries have developed their own versions of the IGB, like the pear-shaped Taylor balloon in the UK and the Heliosphere BAG in France. The FDA’s 2015 approval of newer models like the Reshape Duo and Orbera marked significant advancements in the U.S. market [3].

The placement of an intragastric balloon has several advantages over other weight loss therapies—it is reversible, repeatable, less invasive, and preserves the anatomy of the gastrointestinal tract [1]. Balloons themselves differ as to the method of insertion, filling medium, and recommended implantation duration. They can be used as a standalone method for achieving weight loss in overweight or obese patients or as a bridge therapy before surgical treatment.

The purpose of this study was to evaluate our recent (July 2018–December 2022) findings and data concerning IGB implantations, with emphasis on efficacy as a bridge therapy before LSG in a high-volume bariatric center. Starting with a 6-month treatment, we aimed to evaluate and compare Medsil (6-month), Orbera365 (up to 12-month label), and Spatz3 (up to 12-month label) as bridging devices in super-obese adults (BMI ≥ 50 kg/m^2^).

## 2. Materials and Methods

The data of 148 patients with super-obesity (BMI ≥ 50 kg/m^2^) who underwent IGB implantation as bridging therapy prior to LSG between July 2018 and December 2022 in our department were collected and analyzed. Patients were divided into three groups depending on the type of implanted IGB device: non-adjustable Orbera365 (47 patients, 31.76%), non-adjustable Medsil (53 patients, 35.81%), and adjustable Spatz3 (48 patients, 32.43%). Treatment allocation of a specific IGB device (Orbera365, Medsil, or Spatz3) was not randomized. Instead, assignment was determined by hospital procurement cycles: during a given procurement period, only one balloon type was stocked and implanted, after which the next device became the sole available option. Thus, eligible patients received the balloon that was currently available, without clinician preference or patient selection among devices. During the study, all patients remained under continuous specialist care and were instructed to follow dietary and life-style recommendations. Detailed patient characteristics are presented in Table 1.

Informed consent was obtained from all individual participants included in the study. IGBs were implanted for 6 months as a bridging therapy prior to bariatric surgery. All the procedures were performed under intravenous sedation with midazolam and fentanyl using a standard gastroscope (Pentax EZG29-i10 9.8 mm diameter with 3.2 mm working channel; PENTAX Medical Europe GmbH, Hamburg, Germany). The heart rate and oxygen saturation of the patients were monitored during and for 1 h after the procedure. Primarily, all IGBs were filled with 700 mL of saline with the addition of 2–3 mL of a 1% solution of methylene blue.

Subsequently, all patients were reassessed after 3 months. During this follow-up, the patients in the Spatz3 group received a balloon volume increase procedure to a total of 850 mL (Figure 1). The adjustment was performed on average at week 12 (range 11–13 weeks). Although Orbera365 and Spatz3 IGB have a recommended therapy duration of 12 months, in our patients, they were explanted endoscopically after 6 months (same as Medsil) from the initial procedure, and the data were collected.

This retrospective study was conducted in accordance with the ethical standards of the institutional and national research committee and with the 1964 Helsinki Declaration and its later amendments. Formal approval by the Institutional Review Board (IRB) was waived due to the study’s retrospective design, which involved analysis of fully anonymized clinical data without additional interventions.

Statistical analysis was conducted using STATISTICA 13 (Dell), adopting an alpha level of 0.05, testing normality of data distribution with the Shapiro–Wilk test (none was parametric), and for multiple linear comparisons using the Kruskal–Wallis test with Dunn’s test for post hoc comparisons and Chi^2^ test for nominal comparisons. %EWL was derived as: [(Initial Weight − Current Weight)/(Initial Weight − IBW)] × 100, reflecting weight loss relative to metabolic risk reduction targets.

Because device availability changed in discrete time blocks, we considered the possibility of period (secular) effects. Baseline characteristics were compared across groups to assess comparability of cohorts implanted in different calendar periods. Group comparisons of outcomes were conducted as specified; *p*-values are reported with the corresponding tests.

## 3. Results

No statistically significant differences were found in the selection of groups in terms of age and sex. There were statistically significant differences in BMI reduction, weight loss, %EWL, and %TWL between the groups.

There were no early IGB removals (0/148) and no major IGB-related complications during the 6-month bridging period. Reduction of BMI after 6 months regardless of sex was highest in the Spatz3 group (mean 8 kg/m^2^, median 7.88 kg/m^2^, range 7.06–10.55 kg/m^2^) in comparison with the Medsil (mean 5 kg/m^2^, median 5.17 kg/m^2^, range 4.54–9.54 kg/m^2^) and Orbera365 (mean 5 kg/m^2^, median 4.62 kg/m^2^, range 4.19–6.24 kg/m^2^) groups, *p* < 0.001.

This trend persisted when analyzing male and female subgroups separately. In women, reduction of BMI after 6 months was highest in the Spatz3 group (mean 8.03 kg/m^2^, median 7.79 kg/m^2^, range 7.06–10.55 kg/m^2^) in comparison with the Medsil (mean 5.42 kg/m^2^, median 5.08 kg/m^2^, range 4.54–8.86 kg/m^2^) and Orbera365 (mean 4.74 kg/m^2^, median 4.62 kg/m^2^, range 4.19–6.24 kg/m^2^) groups, *p* < 0.001.

In men, BMI reduction after 6 months was also highest in the Spatz3 group (mean 7.98 kg/m^2^, median 7.95 kg/m^2^, range 7.15–9.35 kg/m^2^) in comparison with the Medsil (mean 5.45 kg/m^2^, median 5.22 kg/m^2^, range 4.57–9.54 kg/m^2^) and Orbera365 (mean 4.61 kg/m^2^, median 4.61 kg/m^2^, range 4.24–4.97 kg/m^2^) groups, *p* < 0.001.

Weight loss after 6 months regardless of sex was highest in the Spatz3 group (mean 25 kg, median 24.64 kg, range 19.46–33.04 kg) in comparison with the Medsil (mean 16 kg, median 16 kg, range 11.7–33 kg) and Orbera365 (mean 14 kg, median 14.53 kg, range 11.54–18.26 kg) groups, *p* < 0.001.

These trends persisted when analyzing male and female subgroups separately. In women, weight reduction after 6 months was highest in the Spatz3 group (mean 24.03 kg, median 25 kg, range 19–31 kg) in comparison with the Medsil (mean 15 kg, median 16 kg, range 12–25 kg) and Orbera365 (mean 14.28 kg, median 14 kg, range 12–18 kg) groups, *p* < 0.001.

In men, weight reduction after 6 months was also highest in the Spatz3 group (mean 25.67 kg, median 26 kg, range 20–33 kg) in comparison with the Medsil (mean 17.75 kg, median 17 kg, range 14–33 kg) and Orbera365 (mean 14.61 kg, median 15 kg, range 12–17 kg) groups, *p* < 0.001.

%EWL after 6 months regardless of sex was greatest in the Spatz3 group (mean 22.98%, median 23.10%, range 19.54–25.53%) in comparison with the Medsil (mean 15.06%, median 14.88%, range 12.49–29.99%) and Orbera365 (mean 13.71%, median 13.88%, range 11.58–15.14%) groups, *p* < 0.001 (Figure 2).

This trend persisted also when analyzing male and female subgroups separately. In women, %EWL after 6 months was highest in the Spatz3 group (mean 22.56%, median 22.61%, range 19.54–24.46%) in comparison with the Medsil (mean 15.01%, median 14.73%, range 12.49–29.99%) and Orbera365 (mean 13.39%, median 13.40%, range 11.58–14.55%) groups, *p* < 0.001.

In men, %EWL after 6 months was also highest in the Spatz3 group (mean 23.67%, median 23.48%, range 19.54–24.26%) in comparison with the Medsil (mean 16.05%, median 15.44%, range 14.21–21.21%) and Orbera365 (mean 14.22%, median 14.09%, range 13.45–15.14%) groups, *p* < 0.001.

To provide a more comprehensive comparison of the IGB therapies, we additionally analyzed the percentage of total weight loss (%TWL) after 6 months (Figure 3). %TWL was highest in the Spatz3 group (mean 14.0%, median 13.9%; range 10.8–18.5%) compared to the Medsil group (mean 9.4%, median 9.2%; range 6.5–14.7%) and the Orbera365 group (mean 8.3%, median 8.1%; range 6.1–11.2%). These findings confirm the superior performance of the adjustable balloon (Spatz3) in terms of overall weight reduction prior to bariatric surgery. The inclusion of %TWL aligns with current recommendations for reporting weight loss outcomes and enhances the clinical relevance of the results.

## 4. Discussion

IGB device implantation can be safely used as a stand-alone procedure as well as a bridge therapy before definitive bariatric surgery. Nevertheless, current knowledge shows that IGB therapy results in more satisfying weight-loss effects in the short term than in long-term follow-up [4]. The best outcomes are achieved with IGB implantation as a bridge therapy for definitive surgical treatment [5,6], and this recommendation is supported in current guidelines [7]. IGB may also be considered as a stand-alone intervention for selected patients not eligible for surgery (e.g., BMI < 40 kg/m^2^ or <35 kg/m^2^ with obesity-related comorbidities) or where access to bariatric surgery is limited or cost-prohibitive.

According to the Spanish Intragastric Balloon Consensus Statement (SIBC) from 2022, IGB should be considered as a therapeutic method in patients with BMI > 25 kg/m^2^ who gain weight despite proper clinical treatment. Therapy should last at least 6 months, and fluid-filled balloons are preferred; no particular brand/type is recommended as superior. The balloon, whether adjustable or not, should be filled with 500–599 mL of fluid with the addition of methylene blue. The decision of changing the IGB volume depends on clinical presentation of the patients—additional volume for diminishing balloon effect/unsatisfactory weight loss or removing volume in case of intolerance. Most often, the volume addition is between 200 and 300 mL, and the reduced volume is 100–150 mL. Some physicians use a syringe manually, and some use a pump, but there is no consensus on this. The overall rate of complications after IGB implantation is 7.07%, of which 0.7% is for minor complications. The intolerance rate is around 5% and results in a 3.62% early removal rate (up to 1 month after implantation) [8]. Approximately 90% of IGB patients experience some form of discomfort—most commonly nausea, vomiting, abdominal pain, or constipation. Thus, patients are treated prophylactically with proton-pump inhibitors and antiemetics before the procedure, as well as antispasmodic drugs during the first week post-insertion [1].

The Brazilian Intragastric Balloon Consensus Statement (BIBC) from 2017 provides a broader indication for IGB implantation when it comes to the age of the patients—in Brazil, it is minimum of 12 years old. Other indications, preparation, the choice of balloon type, technique, and recommendation are similar to the Spanish ones. Authors made a statement that there was no difference in weight loss among different types of IGB [9]. This conclusion was made on the basis of the randomized controlled trial (2010) by De Castro et al., where outcomes from the therapy with two non-adjustable IGBs (Heliosphere and Bioenterics-BIB) were compared [10].

Bridge therapy is recommended in super-obese patients (BMI ≥ 50 kg/m^2^) to reduce the risk of morbidity and mortality following the definitive surgical procedure in this group of patients [2]. In such patients, large fatty livers and enlarged omental fat tissue may present a technical challenge for laparoscopic surgery as well as hinder proper visualization of anatomical structures—rendering successful surgery less likely. In addition, bridge therapy is associated with better blood pressure control, improved glucose metabolism, lower risk of a thromboembolism, lower conversion rate, and shorter surgery as well as hospital stay. These beneficial results may stem from the decreased amount of visceral fat and a decrease in liver volume [2]. This is the so-called “bridge to safe surgery”. According to a study conducted by Ball et al. in 2019, approximately 63% of patients with an IGB inserted prior to proposed bariatric surgery received it [5]. The remainder who did not have a pre-operative IGB did not undergo the planned bariatric surgery, citing refusal of operation, psychological reasons, or other contraindications for surgical treatment [5]. There are also contraindications for IGB for which potential patients should be screened. These include but are not limited to hiatal hernia larger than 5 cm, previous gastric surgery, pregnancy, severe liver disease, alcoholism, coagulopathy, and bleeding lesion of upper GI tract [1,2].

There are many types of IGB available in the market but few scientific studies comparing them. In a meta-analysis of randomized trials comparing fluid-filled and gas-filled IGBs from 2018, Bazerbachi et al. concluded that therapy based on fluid-filled IGBs results in weight loss in 96.8% of patients at 6 months and 96.6% of patients at 12 months, which is superior to gas-filled IGBs [11]. However, fluid-filled balloons have a higher risk of intolerance and early removal compared to gas-filled ones [11]. Swei et al. in 2023 also compared gas-filled (Obalon) and fluid-filled (Orbera) IGB devices in 87 patients—57 and 30, respectively [12]. There was no statistically significant difference in percent total body weight loss (%TWBL) at IGB removal and in 12-month follow-up. Also, in this study, fluid-filled IGB devices were proven to have a higher rate of intolerance resulting in early balloon removal [12]. Kozłowska-Petrickzo et al. in 2023 compared the efficacy of 6-month (Orbera) versus 12-month (Orbera365) IGB therapy, proving that there was no significant difference in the mean %TWL between the groups—it was 15.2% and 15.8%, respectively [13]. Earlier comparative studies did not demonstrate a clear efficacy advantage of adjustable over non-adjustable balloons; however, their designs differed materially from ours. For example, in Genco et al. (2013) [14], only 22.5% of patients with an adjustable device actually underwent a mid-course volume increase, and outcomes were similar to those with non-adjustable balloons. Russo et al. (2017) likewise reported no significant differences, with volume adjustment performed in only ~20% of adjustable-balloon recipients [15]. In contrast, our bridging protocol mandated systematic adjustment in 100% of adjustable-balloon cases at ~3 months, which may explain the greater short-term weight loss observed here. These design differences suggest that fully utilizing the adjustability feature could be key to achieving incremental benefit.

The main measures of IGB therapy effectiveness are weight loss, change in BMI, and %EWL. %EWL is defined as a quotient of the amount of weight loss and the difference between patients’ initial weight and ideal body weight (IBW) expressed as a percentage [16]. Devine’s method was used for IBW evaluation in this study [17]. The American Society for Gastrointestinal Endoscopy (ASGE) and the American Society for Metabolic and Bariatric Surgery (ASMBS) Task Force on Endoscopic Bariatric Therapy (EBT) recommends that primary endoscopic bariatric procedures should achieve at least 15% EWL in 12 months follow-up compared to control group, with statistical significance [16]. As mentioned in the recommendation of the ASGE/ASMBS Task Force on EBT, the expected durability of bridge therapy prior to bariatric surgery is not crucial and should not be unnecessarily lengthened. More important is to achieve optimal weight loss in order to reduce morbidity and mortality following surgery [16].

All patients included in this study underwent LSG after IGB removal and remain under long-term follow-up. The absolute advantage of approximately 9–11 kg greater preoperative weight loss in the adjustable-balloon group over 6 months is potentially meaningful for surgical risk modification. Our dataset focuses on the pre-LSG phase and does not include standardized postoperative metrics; therefore, any inference that this additional weight loss improves peri-operative outcomes (conversion, complications, length of stay) should be regarded as a hypothesis to be tested prospectively. Nonetheless, situating these results within established rationales for preoperative weight reduction supports the plausibility of such benefits.

Our observation of 0/148 early removals is lower than rates commonly reported in large series and consensus statements, where intolerance of ≈5% and early removal of ≈3–5% are described. Several factors may account for this difference, including stringent patient selection and counseling, standardized PPI/antiemetic protocols, and close early follow-up.

### Limitations

This single-center retrospective cohort employed availability-based, time-blocked allocation determined by hospital procurement cycles. Although baseline characteristics were compared, unmeasured confounding including differences in comorbidity burden, patient motivation, and adherence to dietetic counseling may have influenced outcomes and were not systematically quantified.

Peri- and postoperative LSG outcomes and long-term weight trajectories were not assessed, and any clinical implications for surgical risk remain inferential. The sample size, while substantial for a single center, may still be underpowered for rare adverse events and subgroup analyses. Findings pertain to super-obese patients (BMI ≥ 50 kg/m^2^) managed within a specialized pathway and may not generalize to patients with lower BMI or to other settings. Potential device or operator learning-curve effects across calendar periods cannot be entirely ruled out. Finally, multiple statistical comparisons introduce a risk of type I error; where applicable, multiplicity-adjusted post hoc tests were applied, yet residual risk remains.

## 5. Conclusions

In this single-center cohort of super-obese adults (BMI ≥ 50 kg/m^2^) undergoing 6-month bridging therapy before LSG, the adjustable balloon with systematic mid-term volume increase showed greater short-term weight-loss efficacy than non-adjustable devices. To our knowledge, this is the first comparison performed in a dedicated bridging context in which 100% of patients in the adjustable-balloon arm underwent protocolized volume adjustment. These findings pertain to the preoperative, short-term window only and should not be generalized to long-term weight maintenance or to IGB as a stand-alone therapy. Larger, prospective studies are needed to confirm these observations and to determine whether the additional preoperative weight loss translates into improved surgical outcomes.

We believe that these results may shed new light on the effectiveness of bridge therapy in a selected group of bariatric patients and may have an impact on further development of this method. Further prospective randomized studies are necessary to evaluate the clinical usefulness of this report.

## Figures and Tables

**Figure 1 jcm-14-07602-f001:**
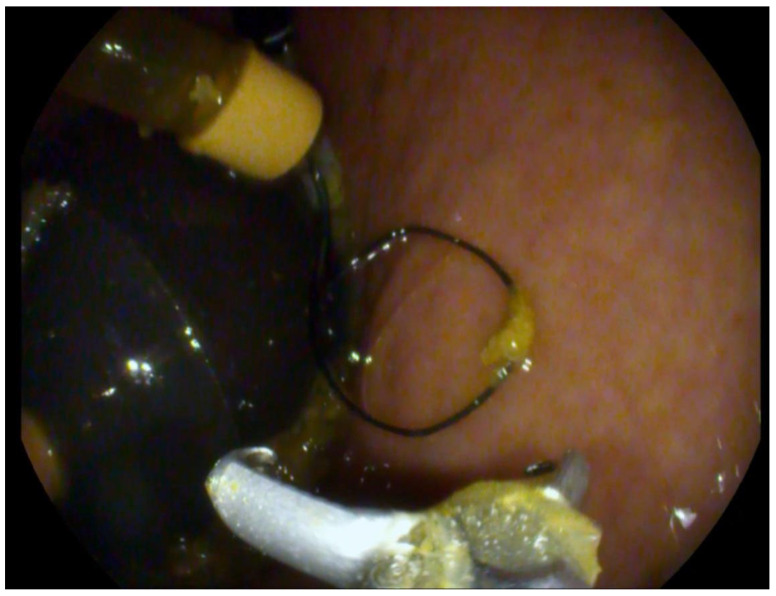
Endoscopic view of Spatz3 regulation catheter during volume adjustment, demonstrating the port used for saline inflation in adjustable IGB therapy.

**Figure 2 jcm-14-07602-f002:**
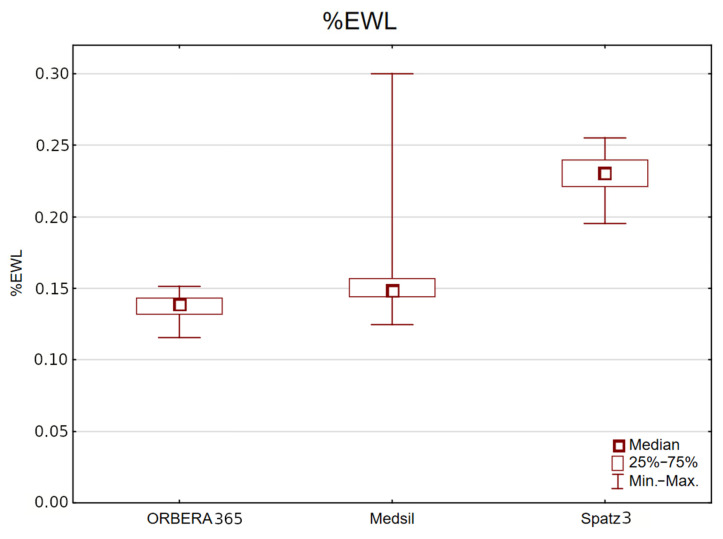
Comparison of %EWL in study groups.

**Figure 3 jcm-14-07602-f003:**
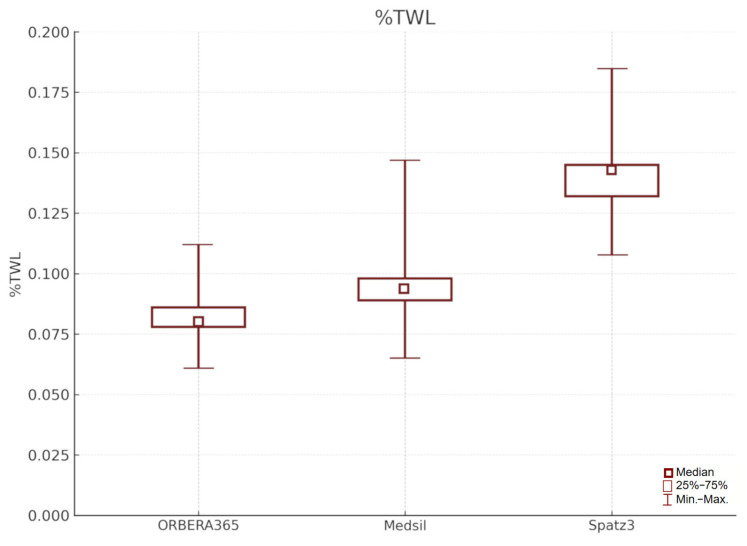
Comparison of %TWL (percentage of total weight loss) in study groups.

**Table 1 jcm-14-07602-t001:** General characteristics of the study groups. Nominal data are presented as n,%. Linear data are presented as min-max, median, and interquartile range. Post hoc p values are listed as follows: for BMI reduction, Medsil vs. Orbera365, *p* > 0.05; Orbera365 vs. Spatz3 = 0.011; Spatz3 vs. Medsil = 0.009; and for %EWL, Medsil vs. Orbera 365, *p* = 0.000; Orbera365 vs. Spatz3 = 0.000; Spatz3 vs. Medsil = 0.000. BMI—body mass index, %EWL—percentage of excess weight loss, %TWL—percentage of total weight loss, IQR—interquartile range.

	Orbera365	Medsil	Spatz3	*p*
Sex (N, %) Male Female	47; 31.71% 18; 38.3% 29; 61.7%	53; 35.81% 20; 37.7% 33; 62.3%	48; 32.43% 15; 31.3% 33; 68.7%	*p* = 0.724
Age (years)	19–56; 33; 26.0–42.5	19–59; 33; 26.0–41.0	19–59; 33; 26.0–42.5	*p* = 0.963
BMI reduction (kg/m^2^)	4.19–6.24; 4.62; 4.40–4.90	4.54–9.54; 5; 4.79–5.58	7.06–10.55; 7.88; 7.32–8.37	*p* < 0.001
Weight loss (kg)	11.54–18.26; 14.53; 13.5–15.0	11.7–33.0; 16; 14.5–17	19.46–33.04; 24.64; 22.0–26.0	*p* < 0.001
%EWL	11.58–15.14; 13.88; 13.0–14.0	12.49–29.99; 14.88; 14.5–15.5	19.54–25.53; 23.10; 21.0–23.0	*p* < 0.001
%TWL	6.1–11.2; 8.1; 7.8–8.6	6.5–14.7; 9.2; 8.9–9.8	10.8–18.5; 13.9; 13.2–14.5	*p* < 0.001

## Data Availability

The data presented in this study are available on request from the corresponding author. The data are not publicly available due to privacy and ethical restrictions.

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
