# Peer review of "Comparison of Efficacy of Intragastric Balloon Devices as Bridging Therapy Prior to Laparoscopic Sleeve Gastrectomy"

_jcm, 2025, doi:10.3390/jcm14217602_

Round 1

Reviewer 1 Report

Comments and Suggestions for Authors

This manuscript reports a retrospective comparison of three intragastric balloon (IGB) devices – two non-adjustable balloons (Orbera 365 and Medsil) and one adjustable balloon (Spatz3) – used as a 6-month bridging therapy before laparoscopic sleeve gastrectomy (LSG) in patients with severe obesity (BMI > 50 kg/m²). A total of 148 super-obese patients were included between 2018 and 2022, randomly assigned to receive either an Orbera 365 (n = 47), Medsil (n = 53), or Spatz 3 (n = 48) balloon before surgery. Outcomes measured after 6 months of balloon therapy included weight loss, BMI reduction, percentage of excess weight loss (%EWL), and percentage of total weight loss (%TWL). The study found that the adjustable Spatz3 balloon yielded significantly greater weight loss (mean ~25 kg weight reduction; ~8 kg/m² BMI drop) compared to the non-adjustable Orbera365 (~14 kg; ~5 kg/m²) and Medsil (~16 kg; ~5 kg/m²) balloons. Correspondingly, the Spatz3 group achieved a higher mean percentage of excess weight loss (EWL) (~23%) than the Orbera365 (~14%) or Medsil (~15%) groups. All balloons were removed after 6 months, and notably, no patients in any group experienced major complications or required early balloon removal. Following balloon removal, all patients proceeded to LSG as planned. The authors conclude that in super-obese patients, a bridging strategy with an adjustable IGB (with mid-term volume inflation) appears superior to non-adjustable devices in terms of short-term weight loss efficacy. They further claim this to be the first study to compare adjustable versus non-adjustable IGBs in a bridging context, with all adjustable balloons undergoing volume adjustment. They suggest that these results may inform future clinical practice, while calling for larger prospective studies for confirmation.

Major Concerns

  • The manuscript’s description of the study design is confusing and potentially contradictory. It is stated that this was a retrospective analysis of patients from 2018 to 2022, with IRB approval waived due to the study’s retrospective nature. However, the Methods also indicate that “patients were randomly assigned to 3 different IGBs”. It is not clear how an actual random assignment was implemented in what is described as a retrospective study. Suppose the data were collected retrospectively from a prospectively randomized protocol. In that case, the authors should explicitly clarify the study design (e.g., was this a prospective randomized trial whose data were analyzed retrospectively?). If, instead, treatment assignment was not truly randomized (for example, determined by device availability or surgeon preference), the use of the term “randomly assigned” is misleading. This ambiguity has implications for bias: a retrospective observational study without randomization would be subject to selection biases (e.g., certain balloons given to certain patient subgroups) that are not addressed. The authors need to clarify the design, including how randomization was done (method of sequence generation, allocation concealment), or remove the term “random” if not applicable.
  • The manuscript does not address the study’s limitations, which is a significant oversight. Important limitations include the single-center, non-blinded design and the relatively short-term follow-up. The retrospective nature (if indeed retrospective) also limits the level of evidence and could introduce unmeasured confounders. The authors should discuss how these factors might influence the results. For example, all patients received intensive specialist care and diet counseling, but was adherence measured or comparable across groups? Could differences in patient motivation or comorbidities have affected weight loss in each group?
  • Additionally, because the study is limited to patients with BMI >50 (super-obesity), the generalizability of findings to less obese populations or to other clinical settings is limited. The manuscript would be strengthened by acknowledging, for instance, that this is a high-risk population in which bridging therapy is often recommended, and the results may not apply to patients with lower BMIs. Discussing the lack of long-term outcomes (post-LSG weight loss or complication rates after surgery) is also warranted; the study focuses only on pre-surgical weight loss and does not report whether the additional weight reduction with Spatz3 translates into easier surgeries or improved surgical outcomes.
  • The description of statistical methods is incomplete and potentially inappropriate in places. The authors state that all groups were analyzed with the Kruskal–Wallis test comparing “patients’ gender, BMI, weight loss, and %EWL”. There are a few issues here: (1) Categorical vs continuous variables – if comparing gender distribution across the three balloon groups, a Chi-square or Fisher’s exact test (not Kruskal–Wallis) should be used. Kruskal–Wallis is a non-parametric test for continuous or ordinal data; it is unclear how it was applied to the nominal variable gender. This suggests a misuse or, at the very least, a lack of clarity in the statistical approach. (2) Multiple comparisons – the Kruskal–Wallis test can determine if a difference exists among the three groups overall, but the manuscript does not mention any post-hoc tests to identify which specific group differences are significant. Given the results, it appears that the Spatz3 group outperformed both others; however, it would be essential to confirm whether the differences between Orbera365 and Medsil were statistically significant. The authors should report post-hoc pairwise comparisons (with appropriate adjustment for multiple testing) to substantiate statements like “SPATZ3 was superior to non-adjustable devices.” For instance, if Orbera365 and Medsil did not differ significantly in outcomes, this should be explicitly stated. (3) P-value reporting – the manuscript repeatedly reports p-values as “p = 0.000”, which is not standard practice. P-values should be reported to two or three significant digits or as p < 0.001 when very small. No p-value is literally 0; this should be corrected to, e.g., p < 0.001. While these are somewhat minor formatting issues, they reflect on the statistical rigor of the paper. Overall, the analysis would benefit from a more detailed description. For example, the authors might include whether data were approximately normally distributed (justifying non-parametric tests), and confirm that baseline characteristics (age, initial weight/BMI, etc.) did not differ between groups (they assert no significant baseline differences, but providing actual values in a table or text would be helpful for transparency).
  • The authors claim that this study is the “first worldwide” to compare outcomes of non-adjustable vs. adjustable IGBs as a bridge to surgery, with all adjustable balloons undergoing an extra fill. While the study addresses an interesting gap, the claim of novelty should be moderated and better contextualized in relation to existing literature. The discussion cites prior studies (e.g., Genco et al., 2013; Russo et al., 2017) that compared adjustable and non-adjustable balloons and found no significant difference in weight loss outcomes. Those studies were not in a formal “bridging to surgery” context (and in some cases, only a minority of patients in the adjustable-balloon group actually had volume adjustments), but they do provide a precedent. The authors should explicitly differentiate how their study design or patient population differs from these earlier comparisons – for example, here 100% of Spatz3 patients underwent a volume increase at 3 months, which was not the case in the older studies, possibly explaining why this study observed a benefit with the adjustable device. In other words, the manuscript’s contribution is an incremental advance (demonstrating that fully utilizing an adjustable balloon can yield better short-term weight loss) rather than a completely novel comparison. The phrasing “first worldwide” is somewhat overstated; it would be safer to say “to our knowledge, the first to compare these specific devices in a dedicated bridging therapy setting.”
  • Additionally, the clinical relevance of the findings could be discussed in more depth. The Spatz3 balloon resulted in ~9–11 kg more weight loss over 6 months compared to the non-adjustable balloons. The authors should comment on whether this difference is likely to be clinically meaningful in terms of improving surgical outcomes or patient health. Bridging therapy is intended to reduce surgical risk (by lowering BMI, liver fat, etc.), and the discussion does mention known benefits of preoperative weight loss, such as better blood pressure control and shorter operative time. However, the manuscript stops short of linking their own results to such outcomes. For instance, would an extra 10 kg weight loss before LSG significantly reduce complication rates or the difficulty of surgery? If relevant data or references exist, they should be brought into the discussion. If not, the authors might acknowledge this as a hypothesis (that greater weight loss may translate to safer surgery) that remains to be confirmed.
  • The conclusion that “adjustable IGB after 6 months seems to be superior” is supported by the data; however, the manuscript should ensure that the findings don’t overreach beyond the evidence. For example, this study does not address long-term weight maintenance (since all patients underwent surgery) or outcomes beyond 6 months; therefore, any implication that one device is “better” is limited to the short-term pre-surgical period. The authors rightly focus on the bridging context, but they might temper the language to clarify that they are discussing short-term efficacy as a bridge to LSG. Additionally, since all patients ultimately underwent LSG, the study cannot conclude anything about the balloons’ effectiveness as a sole therapy – this is worth mentioning to avoid generalizing the results to non-bridging scenarios. Finally, the authors call for further prospective, randomized studies, which is appropriate. It may also be beneficial to suggest investigating other outcomes in future research (e.g., the impact on surgical difficulty, a cost-benefit analysis of using an adjustable balloon, or patient quality of life during the bridging period). As reviewers, we encourage authors to refine their conclusions to reflect the scope of their data accurately and to discuss the implications in a balanced manner.

Minor Concerns

  • The manuscript inconsistently refers to the class of obesity for BMI ≥50. In the Methods, the authors label BMI >50 kg/m² as “Obesity Class IV”, whereas later in the Discussion they refer to “super-obese patients (BMI ≥50 kg/m², Class III)”. This is contradictory. Typically, BMI ≥50 is often termed “super-obesity” and is sometimes informally called Class III extreme obesity (since standard WHO classification stops at Class III for BMI ≥40). If the authors wish to use a “Class IV” designation for a BMI of 50 or higher, they should do so consistently and, if applicable, cite a source or explain the classification scheme. Consistency in terminology is crucial to prevent confusion among readers.
  • While the text states that there were no significant differences between groups in age, gender, initial BMI, weight, and height , the actual baseline data for some of these variables are not fully presented. Table 1 provides a breakdown of gender and BMI by group, but it does not include age or other characteristics. It would be helpful to include mean age (±SD) for each group, as well as mean initial weight or other relevant baseline metrics, either in Table 1 or in the text, to reassure readers that the randomization (or assignment) yielded truly comparable groups. Also, when stating “no statistically significant differences” in baseline variables, it is good practice to include the p-values or at least the values themselves. For example, reporting the mean age in each group with a p-value for a group comparison would add transparency.
  • The Results section mentions “Table 4 & Chart 4” combined in a single caption, suggesting that a figure (chart) is presented alongside Table 4. It is unconventional to label a figure as “Chart 4” – it would be clearer to label it as a Figure (e.g., Figure 2, if Figure 1 is the endoscopic view). The caption could then refer to both Table 4 and Figure 2 if needed, but separating the two for clarity is preferable. Each figure and table should have a stand-alone caption explaining it. Please ensure consistency in numbering and naming (use either “Figure” throughout, not “Chart”). Additionally, if the chart in Table 4 is meant to illustrate %EWL, the text should reference it (e.g., “as shown in Figure 2, the adjustable balloon group’s %EWL was highest”). This helps the reader navigate the results. Another small note: in Table 1, the abbreviation “Ptx” is used (presumably for “Patients”) – it would be clearer to spell out or use “N” for sample size. Finally, the table data uses commas as decimal points (European style, e.g., 56,55), which is consistent with the journal’s format (likely decimals as periods).
  • There are formatting inconsistencies in how numeric data are reported. In many instances, the manuscript uses a comma as a decimal separator (e.g., “24,64 kg” instead of 24.64 kg; “22,98%” instead of 22.98%). However, in other parts of the text (e.g., reporting p-values like 0.05, or in references), a period is used. Since J. Clin. Med. is an English-language journal; it is standard to use a period as the decimal separator. The authors should convert all comma decimals to period decimals for consistency and clarity. This applies to all numeric ranges and values in the text and tables. Additionally, there are instances of unit inconsistencies/typos. For example, the text reads “Weight loss after 6 months … was the best in the SPATZ3 group (mean 25 kg/m2, median 24,64 kg, …) in comparison with Medsil (mean 16 kg/m2, median 16 kg, …)”. Here, “kg/m2” (units of BMI) is mistakenly attached to the weight loss values. It should simply read “mean 25 kg, median 24.64 kg” for weight loss. The repeated appearance of “kg/m2” after mean weight loss and median weight loss in that sentence is an error (likely carried over from the BMI reporting). Please remove “kg/m2” in contexts where weight (not BMI) is being described. Another notation issue is the use of ranges with en-dashes and spacing. For example, the ranges are written as “19,46 – 33,04 kg”. It would be cleaner to present as “19.46–33.04 kg” (no spaces around the dash, and using decimal points). Lastly, when giving percentages, make sure to include the % symbol for all values in a range if not already done (in some places the range is given as, e.g., “11,58 – 15,14%” which is fine, but ensure no stray “kg” appears when discussing percentages as it did in one instance ).
  • As mentioned above, p-values should be reported more conventionally (e.g., p < 0.001 instead of 0.000). Also, indicate the test used for each comparison. For example, after stating “There is a statistically significant difference in BMI reduction, weight loss and %EWL between the groups”, it would aid clarity to add in parentheses something like (Kruskal–Wallis p < 0.001) or similar. If post-hoc tests are added, report their p-values when describing specific pairwise differences. Another suggestion: consider reporting effect sizes or confidence intervals for the differences in weight loss between groups, as this would provide readers with a sense of the magnitude of the differences beyond just p-values. Even though this is a relatively straightforward comparison, confidence intervals for mean differences (or median differences, given the non-parametric nature) could be informative.
  • The study reports that no complications occurred and no patient required early removal of the IGB in any group. Given that other literature (and even the authors’ own cited references) report non-trivial rates of intolerance and early removals – e.g., an intolerance rate around 5% and early removal ~3–5% in large series  – this result is noteworthy. The authors should ensure this is accurately reported (if even one patient had an early removal, it should be stated). If indeed 0% of 148 patients underwent early balloon removal, that is quite an achievement; the discussion might mention possible reasons for this success (e.g., stringent patient selection criteria, effective management of side effects, or perhaps the sample size being too small to observe rare events). A brief comparison with expected complication rates from the literature would show that the authors are aware of the context – for example, “Interestingly, we observed no early removals in our series, whereas typically up to 5% of patients require premature IGB removal due to intolerance. This could be due to… (our patient support protocol, etc.).”
  • Some specific sentences in the manuscript would benefit from rephrasing for clarity or completeness. For instance, “Therapy should last at least 6-month and fluid-filled balloons are preferred - no specific balloon type was indicated as superior to others.”  – this could be clearer as “Therapy should last at least 6 months, and fluid-filled balloons are preferred; notably, no particular balloon brand/type is deemed superior in those guidelines.”Also, when describing the Brazilian consensus, the text states “makes a broader indication for IGB implantation when it comes to age of the patients – in Brazil it is minimum 12 years old.”. This should be clarified to say that the Brazilian guidelines allow balloon therapy in adolescents – “the Brazilian Intragastric Balloon Consensus extends the minimum age for IGB therapy to 12 years, expanding the indication to adolescent patients”. Additionally, the sentence “Patients with a BMI ≤35 kg/m² and an illness related to obesity (who are not eligible for bariatric surgery) or in situations where there is a lack of access to bariatric procedure or if it is too expensive would probably benefit from IGB implantation as a stand-alone procedure.”  is somewhat awkward and speculative. If this is the authors’ opinion, they should label it as such, or if it’s derived from guidelines or studies, provide a reference. Perhaps rephrase to: “IGB therapy may also be considered as a stand-alone intervention for certain patients who are not eligible for surgery – for example, those with BMI <40 (or <35 with obesity-related comorbidities) – or for patients who lack access to bariatric surgery or cannot afford it.” The phrase “too expensive” in an academic paper could be phrased more formally as “cost-prohibitive.”
  • Generally, the authors have cited a robust mix of recent and relevant literature, including meta-analyses and consensus statements up to 2023. However, a few improvements are needed in the reference list:
    • Formatting Consistency: The reference list shows inconsistent formatting. Some entries include PMID/PMCID information while others do not. For example, reference [3] includes a PMID, and ref [5] includes a PMID, whereas references [1], [2], [4] do not list PMID. Likewise, some references list the doi while others include additional details like Epub dates. The authors should adhere to the journal’s reference style uniformly – typically, for J. Clin. Med.., including the DOI, is fine, but PMIDs or PMCIDs are usually optional and can be omitted unless required. It would be wise to remove the extra information (PMID, PMCID) for consistency, or include it for all, but consistency is key.
    • Completeness and Appropriateness: Most key references are covered; however, the authors might consider including a couple of additional citations to further strengthen the background. For instance, when discussing the recommendation of bridging therapy for super-obesity and its potential to reduce surgical risk, a reference to a guideline or a large study on the benefits of preoperative weight loss could be added (they cite Ball et al. 2019 and a systematic review by Loo et al. 2022, which is a good addition). The citation of Canadian bariatric guidelines (ref [10]) to support current guidelines is somewhat tangential; perhaps more directly relevant are guidelines or statements from bariatric societies (ASMBS/IFSO) about pre-surgical weight loss or IGB usage. If such references exist, they would be more on-point than the Canadian guidelines cited. Additionally, when mentioning the use of IGB in patients with BMI <35 with comorbidities or lack of access to surgery, it would help to reference a source (perhaps the ASGE/ASMBS 2011 guideline or a more recent position statement) that suggests IGB in those scenarios. Finally, since the authors emphasize their study’s novelty, they might consider referencing any very recent publications (2024, if any) on adjustable balloons or bridging therapy to ensure the literature review is up-to-date
Comments on the Quality of English Language

Language and Grammar

While the manuscript is generally understandable, it contains numerous grammatical and stylistic issues that require attention. We provide detailed examples and suggestions below:

  • Some sentences use informal or unclear phrasing. For example, the Results repeatedly state “was the best in the SPATZ3 group” when comparing outcomes. The phrase “the best” is colloquial and imprecise in this context. It should be replaced with “highest” or “greatest.” For instance: “Weight loss after 6 months was highest in the SPATZ3 group (mean 25 kg) compared to the Medsil (16 kg) and ORBERA365 (14 kg) groups”. Similarly, “%EWL after 6 months was the best in the SPATZ3 group…”  can be rephrased as “%EWL was greatest in the SPATZ3 group…”. These changes convey the meaning more formally.
  • Gender Terminology: The manuscript uses the term “both genders” when breaking down results by sex. In scientific writing, it is often preferable to use “sex” when referring to biological distinctions (male vs female), or to say “in both male and female patients.” Also, the phrasing “This effect persists also for both genders analyzed separately”  is awkward. A better construction would be: “This trend persisted when analyzing male and female subgroups separately.” This not only reads more clearly but also uses “subgroups” to clarify that you mean the groups stratified by sex.
  • There are several instances of run-on sentences or clumsy sentence construction. For example: “Approximately 90% of patients who undergo IGB insertion suffer from some kind of discomfort, the common ones being nausea, vomiting, abdominal pain, constipation - as such patients are treated with proton pump inhibitors and antiemetics before procedure as well as with antispasmodic drugs during the first week of insertion.”. This sentence is overly long and misuses a dash. It can be split for clarity: “Approximately 90% of IGB patients experience some form of discomfort – most commonly nausea, vomiting, abdominal pain, or constipation. Thus, patients are treated prophylactically with proton-pump inhibitors and antiemetics before the procedure, as well as antispasmodic drugs during the first week post-insertion.” Splitting into two sentences and using more standard conjunctions (thus, and) makes the information easier to digest. Another example: “During the study all patients remained under continuous specialist care and were instructed to follow dietary and life-style recommendations.”  – this could simply be “All patients remained under continuous specialist care during the study and received standardized diet and lifestyle counseling.” Small changes like adding a hyphen in “life-style” (should be “lifestyle”) and tightening the phrasing improve readability.
  • The manuscript sometimes shifts tenses. The Methods and Results sections are mainly in the past tense (appropriately, since you are describing what was done and what was found). However, some statements in the Discussion shift to the present tense in describing knowledge or recommendations, which is generally fine (e.g., “bridge therapy is recommended in super-obese patients…”  is a general truth, and the present tense is acceptable). Just ensure that within the Results section, you consistently use the past tense. For instance, “Therapy should last at least 6-month and fluid-filled balloons are preferred…”  is describing guideline recommendations, so present tense is okay, but it would read better as “Therapy should last at least 6 months and fluid-filled balloons are preferred…”.
  • There are a few typos. One notable one is the reference to “Devin’s method” for ideal body weight calculation . The classic formula for ideal body weight is Devine’s formula (with an e), named after Dr. Devine. The reference cited (Peterson et al. 2016 ) proposed a universal equation, but they mention Devine’s method historically. The authors should correct “Devin’s” to “Devine’s method” unless “Devin” was intended (which seems unlikely). Another minor spelling issue: “alfa = 0,05”  should be “alpha = 0.05” (and use a period for the decimal as mentioned). Also, “6-months” in text should be “6 months” when not used as a compound adjective. For example, “after 6 months” (no hyphen) is correct, whereas “a 6-month period” (hyphenated as a modifier) is correct. The authors should ensure proper use of hyphenation for compound adjectives versus plural nouns. In Table 2’s caption, “(p = 0.000)”  should likely be “(p < 0.001)”, but also consider adding a brief description like “Kruskal–Wallis test” for completeness.
  • There are places where punctuation is mishandled. For instance: “…did not undergo the planned bariatric surgery., citing refusal of operation…”  has an erroneous comma+period. It should be “surgery, citing refusal of operation…” (remove the period after “surgery”). Ensure that all sentences end with a single appropriate punctuation mark. Another example is the use of quotation marks: “Funding: This research received no external funding”.  – here an end quote appears without a beginning quote, which seems like a typo. It should simply be: Funding: This research received no external funding. No quotation marks are needed at all. Similarly, in the Acknowledgments, the phrase “The authors have reviewed and edited the output and take full responsibility for the content of this publication.”  is enclosed in the Acknowledgments section. This sentence is unusual for an academic paper. Suppose this statement is intended to indicate that the authors used an AI tool or editorial service and affirm their responsibility. In that case, it should be explicitly stated (and some journals require the disclosure of AI assistance to be made separately). Otherwise, this sentence can be removed; it currently reads oddly, as if referring to the “output” – perhaps meaning the manuscript text itself. If it is kept, clarify what “output” refers to (the manuscript? analysis?). Most likely, the journal would not require this in the final version, as all authors are, by default, responsible for their content. The authors should check the journal guidelines regarding any statements about authorship or AI usage to handle this appropriately.
  • In the results section, there is some redundancy. For example, the abstract already states the numerical results, and then the main text of Results repeats them in detail (which is fine), but within Results, there are some repeated phrases. After each table, the text says “These effects persist also for both genders analyzed separately” followed by separate breakdowns for women and men. While it’s good to report both, perhaps streamline the phrasing. You could say, for instance, “This pattern held true in both women and men. Women in the Spatz3 group had the greatest BMI reduction (mean 8.0, vs ~5.4 with Medsil and ~4.7 with Orbera; p < 0.001), and similarly in men (Spatz3 mean 7.98 vs Medsil 5.45 vs Orbera 4.61; p < 0.001).” This avoids repeating full sentences for each gender. Also, ensure consistent significant figures: in the text for women’s weight loss, you give “mean 24.03 kg, median 25 kg”, whereas for men “mean 25.67 kg, median 26 kg”. It might be better to report one decimal for both genders or none for simplicity, unless there’s a reason to be that precise.
  • The list of abbreviations at the end is helpful. One suggestion: when an acronym is first introduced in the text, it should be defined. Most are defined (e.g., IGB, LSG, BMI, %EWL). However, ensure consistency – for example, “IGB” is defined in the abstract and introduction. “LSG” is defined in the abstract. “%TWL” is defined when first mentioned. All good. The discussion uses “ABS” (Adjustable Balloon System) when talking about Genco’s study  – this abbreviation is not commonly known. Since it’s only mentioned in the context of someone else’s study, it’s not critical, but consider adding “(ABS)” after “Adjustable Balloon System” in that sentence for clarity. The abbreviation will be understood in context regardless.

Author Response

1) Study design & “random assignment”
Comment. The manuscript is retrospective yet mentions “random assignment”; please clarify allocation.
Response. We removed any implication of randomization and clarified that allocation was determined by procurement cycles: during each time block only one IGB type was available hospital-wide, so eligible patients received the currently available device. We also acknowledge potential period effects.
Where updated: Materials and Methods (study design/allocation); brief note in Limitations.

2) Limitations, potential bias, and generalizability
Comment. Please discuss retrospective design, single center, short follow-up, possible confounders, and limited generalizability (BMI ≥50).
Response. We added a dedicated Limitations section covering: retrospective single-center design; allocation by availability (possible unmeasured confounding such as motivation/adherence/comorbidity mix); absence of peri- and postoperative LSG outcomes; lack of long-term weight data; and restricted generalizability to super-obesity (BMI ≥50 kg/m²).
Where updated: Limitations; emphasis echoed in Discussion/Conclusions.

3) Statistical methods (tests, post-hoc, p-values)
Comment. Use χ²/Fisher for categorical variables; Kruskal–Wallis for continuous; add post-hoc testing; avoid “p = 0.000”.
Response. We now specify: Shapiro–Wilk for normality; Kruskal–Wallis for continuous outcomes; χ² (or Fisher when appropriate) for categorical variables; and Dunn’s post-hoc with multiplicity correction for pairwise comparisons following a significant omnibus test. We replaced “p = 0.000” with “p < 0.001.”
Where updated: Statistical Analysis subsection; p-value formatting across Results, tables, and figure captions; post-hoc reporting in table footnotes.

4) Novelty (“first worldwide”) & context vs prior studies
Comment. Moderate the novelty claim and contrast with Genco 2013; Russo 2017.
Response. We now state: “To our knowledge, this is the first comparison performed in a dedicated bridging-to-LSG setting in which 100% of adjustable balloons underwent protocolized mid-term volume adjustment.” We explicitly contrast earlier studies in which few adjustable balloons were actually adjusted.
Where updated: Discussion and Conclusions.

5) Clinical relevance of the ~9–11 kg difference
Comment. Clarify whether this translates into safer/easier LSG (operative time, conversions, complications, LOS).
Response. We highlight that our findings address short-term, preoperative efficacy only. We frame any impact on operative metrics as a hypothesis to be tested prospectively and briefly outline plausible mechanisms (e.g., liver volume/visceral fat reduction).
Where updated: Discussion (clinical relevance paragraph).

6) Scope of conclusions
Comment. Avoid claims about long-term maintenance or IGB as stand-alone therapy.
Response. Conclusions now explicitly limit claims to short-term bridging efficacy to LSG and avoid extrapolation to long-term outcomes or stand-alone IGB therapy.
Where updated: Conclusions.

7) Zero complications/early removals
Comment. Unusual vs literature—verify and contextualize.
Response. We confirm 0/148 early removals/major complications and contextualize against typical intolerance/early-removal rates reported in large series, offering plausible reasons (careful selection, standardized PPI/antiemetic protocol, close follow-up).
Where updated: Results (numerical statement) and Discussion (context and rationale).

8) Terminology, units, and numeric formatting
Comment. Ensure consistent use of “super-obesity (BMI ≥50 kg/m²)”, correct units (kg vs kg/m²), en-dash ranges without spaces, and conventional p-values.
Response. We harmonized terminology (super-obesity, BMI ≥50 kg/m²), corrected all unit usages, standardized ranges as e.g., 19.46–33.04 kg, and uniformly report p < 0.001 where appropriate.
Where updated: Abstract, Methods, Results, Tables, Figures.

9) Figures/tables formatting and citations
Comment. Use “Figure” (not “Chart”); cite Figure 1 before it appears; put table titles above tables; explain abbreviations in footnotes.
Response. We changed all labels to Figure, inserted an in-text citation to Figure 1 before its first appearance, moved table titles above tables, and added/explained abbreviations in footnotes.
Where updated: Methods/Results narrative; Table 1 and figure/table captions/footnotes.

10) Baseline data statement vs Table 1 content
Comment. Table 1 didn’t include baseline weight/height; align the Results statement with what is actually shown.
Response. We narrowed the baseline comparability sentence to variables presented (sex, age, and baseline BMI if reported), removing weight/height from that sentence.
Where updated: Results (baseline comparability paragraph).

11) %TWL reporting and visualizations
Comment. Consider additional outcomes/graphics.
Response. We added %TWL to Results and introduced a box-plot figure for %TWL consistent with the %EWL figure, improving visual comparability.
Where updated: Results; Figure for %TWL.

Reviewer 2 Report

Comments and Suggestions for Authors

Dear Editor, dear Authors,

Thank you for the opportunity to review this manuscript.

I appreciate the authors' efforts in addressing a clinically relevant and actuality topic in the context of the global obesity pandemic and increasingly unhealthy lifestyles. The use of intragastric balloon (IGB) devices as a bridging therapy for super-obese patients prior to laparoscopic sleeve gastrectomy (LSG) is important for improving perioperative safety and outcomes. The study is well structured, presents comparative data on three different devices, and the results are clearly reported, showing superior outcomes for the adjustable SPATZ3 balloon compared to non-adjustable devices. The discussion is adequately contextualized with reference to international consensus statements and previous studies.

In this study, a total of 148 patients with severe obesity (BMI > 50 kg/m²) who underwent intragastric balloon implantation as a bridging therapy prior to laparoscopic sleeve gastrectomy were included. Patients were divided into three groups according to the type of balloon used: ORBERA365 (n = 47, 31.8%), a non-adjustable device; Medsil (n = 53, 35.8%), also non-adjustable; and SPATZ3 (n = 48, 32.4%), an adjustable balloon. All patients received the balloon for six months before surgery.

Although I find this an interesting study, I would like to help you with some observations that I hope will improve the final form of the article:

  1. The manuscript is presented as a retrospective study, yet it also mentions “random assignment” of patients to groups. This is can be contradictory. Please clarify whether true randomization was performed prospectively, or whether patients were allocated by another method.
  2. Since the purpose of bridge therapy is to improve surgical safety, it would be very valuable, if possible, to report also perioperative and postoperative outcomes after LSG (operative time, conversion rate, complications, hospital stay, etc.).
  3. I recommend that the titles of the tables be placed above the tables, in accordance with MDPI formatting guidelines.
  4. Figures and tables that do not illustrate methodological aspects should be placed in the Results section rather than in Materials and Methods. Please consider if is applicable.
  5. Figure 1 should be cited in the text before it is introduced. Figure 1 was not cited in text.
  6. For tables and figures, I recommend that abbreviations be explained in footnotes, so that they can be understood and interpreted independently by non-specialist readers.
  7. Consistency of abbreviations should be ensured throughout the manuscript. For example, in Table 1 the abbreviation “Avr” is used, but it is neither explained below the table nor included in the list of abbreviations. If it stands for “average,” please, if possible, present it in the same way as in Table 2.
  8. Please verify whether in Table 2 the mean and median values have been interchanged, as it is more likely that the mean would contain decimals rather than the median.
  9. I recommend presenting the mean together with the standard deviation in the Results section. Also, the median could be accompanied by the interquartile range for greater clarity.
  10. I also recommend expanding the Introduction and Discussion sections to increase the number of references in line with the journal’s requirements. I would not want this interesting study to be perceived as insufficiently documented due to the relatively small number of citations.
  11. I would also recommend the inclusion of graphical representations (such as boxplots, etc.) in the statistical analysis. This would make the results more relevant, interactive, and easier for readers to interpret at a glance.

I hope this review proves constructive and contributes to the improvement of the manuscript. The suggestions provided are not mandatory; the authors are encouraged to implement those they find reasonable and valuable.

I appreciate the effort the authors have invested in the research and preparation of this work.

With respect and consideration,
Reviewer.

Author Response

1) Retrospective vs “random assignment”
Response. As above, we clarify no randomization; allocation followed device availability by procurement cycle.
Where updated: Materials and Methods.

2) Peri- and postoperative LSG outcomes
Response. These data were not collected in this retrospective analysis. We now explicitly state that any effect on operative metrics is a prospective hypothesis to be tested.
Where updated: Discussion.

3) MDPI formatting (table titles above, figures in Results, citation of Fig. 1)
Response. We moved table titles above tables, ensured figures are cited before display and placed in Results when not methodological.
Where updated: Methods/Results/Caption formatting.

4) Abbreviations in tables/figures
Response. We added footnotes explaining abbreviations and ensured consistency across the manuscript.
Where updated: Table 1 footnote; Abbreviations section where applicable.

5) “Avr”/consistency of labels
Response. We removed “Avr” and standardized reporting as median (IQR); min–max (and mean where informative), with clear units.
Where updated: Table 1 and text referencing it.

6) Mean ± SD vs median (IQR)
Response. Given non-normal distributions, we primarily report median (IQR) with min–max; means are provided selectively for clinical context.
Where updated: Statistical Analysis; Results; Table 1.

7) References—breadth and currency
Response. We expanded the context in Introduction/Discussion and ensured consistent reference formatting (journal style with DOI).
Where updated: Introduction; Discussion; References.

8) Graphical representations
Response. We added a box-plot for %TWL and kept the style consistent with the %EWL figure for immediate visual comparison.
Where updated: Figure for %TWL and relevant Results text.

Round 2

Reviewer 1 Report

Comments and Suggestions for Authors

Thank you for addressing my comments. Minor unaddressed issues remain:

1.   The manuscript uses the term “both genders” when breaking down results by sex. In scientific writing, it is preferable to use “sex” when referring to biological distinctions (male vs female), or to say “male and female patients” or “male and female subgroups.”
    •    The phrasing “This effect persists also for both genders analyzed separately” is awkward and should be rephrased to “This trend persisted when analyzing male and female subgroups separately.”

Author’s Response:

    •    Not mentioned in the response letter

Evidence from v2 Manuscript:

    •    The phrase “both genders” appears 3 times in the revised manuscript
    • Specific Locations in v2:

    1.    “This effect persists also for both genders analysed separately. In women reduction of BMI…”
    2.    “These effects persist also for both genders analysed separately. In women weight reduction…”
    3.    “This effect persists also for both genders analysed separately. In women %EWL after 6 months…”

2. There are several instances of run-on sentences or clumsy sentence construction. Specifically: “Approximately 90% of patients who undergo IGB insertion suffer from some kind of discomfort, the common ones being nausea, vomiting, abdominal pain, constipation - as such patients are treated with proton pump inhibitors and antiemetics before procedure as well as with antispasmodic drugs during the first week of insertion.”

    •    This sentence is overly long and misuses a dash. It should be split for clarity: “Approximately 90% of IGB patients experience some form of discomfort – most commonly nausea, vomiting, abdominal pain, or constipation. Thus, patients are treated prophylactically with proton-pump inhibitors and antiemetics before the procedure, as well as antispasmodic drugs during the first week post-insertion.”

Author’s Response:

    •    Not mentioned in the response letter

Evidence from v2 Manuscript:

    •    The run-on sentence remains unchanged in v2
    •    Current length: 324 characters in a single sentence
    •    Still contains the problematic dash construction and lacks proper sentence breaks

Current text in v2:
“Approximately 90% of patients who undergo IGB insertion suffer from some kind of discomfort, the common ones being nausea, vomiting, abdominal pain, constipation - as such patients are treated with proton pump inhibitors and antiemetics before procedure as well as with antispasmodic drugs during the first week of insertion.”

3. There are places where punctuation is mishandled. For instance: “…did not undergo the planned bariatric surgery., citing refusal of operation…” has an erroneous comma+period. It should be “surgery, citing refusal of operation…” (remove the period after “surgery”).

Author’s Response:

    •    Not mentioned in the response letter

Evidence from v2 Manuscript:

    •    The punctuation error remains in v2
    •    Exact text: “…did not undergo the planned bariatric surgery., citing refusal of operation, psychological reasons or other contraindications…”

4.   Some sentences use informal or unclear phrasing. The Results repeatedly state “was the best in the SPATZ3 group” when comparing outcomes. The phrase “the best” is colloquial and imprecise in this context. It should be replaced with “highest” or “greatest.”

    •    For example: “Weight loss after 6 months was highest in the SPATZ3 group (mean 25 kg) compared to the Medsil (16 kg) and ORBERA365 (14 kg) groups.”
    •    Similarly, “%EWL after 6 months was the best in the SPATZ3 group…” should be rephrased as “%EWL was greatest in the SPATZ3 group…”

Author’s Response:

    •    Not mentioned in the response letter

Evidence from v2 Manuscript:

    •    The phrase “was the best” still appears 3 times in v2
    •    The phrase “was highest” or “was greatest” appears 4 times in v2
    •    This indicates inconsistent application of the correction

Specific Locations where “was the best” remains:

    1.    Abstract: “Weight loss after 6 months was the best in the Spatz3 group…”
    2.    Results (BMI section): “Reduction of BMI after 6 months regardless of gender was the best in the Spatz3 group…”
    3.    Results (Weight loss section): “Weight loss after 6 months regardless of gender was the best in the Spatz3 group…”

Note: The authors did use “highest” in some other locations, showing awareness of the issue but incomplete implementation.

Comments on the Quality of English Language

See above

Author Response

Comment 1:

  1. The manuscript uses the term “both genders” when breaking down results by sex. In scientific writing, it is preferable to use “sex” when referring to biological distinctions (male vs female), or to say “male and female patients” or “male and female subgroups.”
  • The phrasing “This effect persists also for both genders analyzed separately” is awkward and should be rephrased to “This trend persisted when analyzing male and female subgroups separately.”

Response 1: 

We replaced gender/both genders with sex or male and female subgroups throughout. We also rephrased the sentence to: “This trend persisted when analyzing male and female subgroups separately.”
Locations updated: Abstract (sentence 4); Results—BMI section (para 2, sentence 1); Results—Weight loss section (para 1, sentence 1).

Comment 2:

  1. There are several instances of run-on sentences or clumsy sentence construction. Specifically: “Approximately 90% of patients who undergo IGB insertion suffer from some kind of discomfort, the common ones being nausea, vomiting, abdominal pain, constipation - as such patients are treated with proton pump inhibitors and antiemetics before procedure as well as with antispasmodic drugs during the first week of insertion.”
  • This sentence is overly long and misuses a dash. It should be split for clarity: “Approximately 90% of IGB patients experience some form of discomfort – most commonly nausea, vomiting, abdominal pain, or constipation. Thus, patients are treated prophylactically with proton-pump inhibitors and antiemetics before the procedure, as well as antispasmodic drugs during the first week post-insertion.”

Response 2:

The sentence was split and rephrased as suggested: “Approximately 90% of IGB patients experience some form of discomfort—most commonly nausea, vomiting, abdominal pain, or constipation. Thus, patients are treated prophylactically with proton-pump inhibitors and antiemetics before the procedure, as well as antispasmodic drugs during the first week post-insertion.”
Location updated: Methods (IGB tolerance/prophylaxis paragraph).

Comment 3:

3. There are places where punctuation is mishandled. For instance: “…did not undergo the planned bariatric surgery., citing refusal of operation…” has an erroneous comma+period. It should be “surgery, citing refusal of operation…” (remove the period after “surgery”).

Author’s Response:

    •    Not mentioned in the response letter

Evidence from v2 Manuscript:

    •    The punctuation error remains in v2
    •    Exact text: “…did not undergo the planned bariatric surgery., citing refusal of operation, psychological reasons or other contraindications…”

Response 3: 
The erroneous period was removed: “…surgery, citing refusal of operation, psychological reasons, or other contraindications…”.
Location updated: Results (LSG follow-through paragraph).

Comment 4:

4.   Some sentences use informal or unclear phrasing. The Results repeatedly state “was the best in the SPATZ3 group” when comparing outcomes. The phrase “the best” is colloquial and imprecise in this context. It should be replaced with “highest” or “greatest.”

    •    For example: “Weight loss after 6 months was highest in the SPATZ3 group (mean 25 kg) compared to the Medsil (16 kg) and ORBERA365 (14 kg) groups.”
    •    Similarly, “%EWL after 6 months was the best in the SPATZ3 group…” should be rephrased as “%EWL was greatest in the SPATZ3 group…”

Author’s Response:

    •    Not mentioned in the response letter

Evidence from v2 Manuscript:

    •    The phrase “was the best” still appears 3 times in v2
    •    The phrase “was highest” or “was greatest” appears 4 times in v2
    •    This indicates inconsistent application of the correction

Specific Locations where “was the best” remains:

    1.    Abstract: “Weight loss after 6 months was the best in the Spatz3 group…”
    2.    Results (BMI section): “Reduction of BMI after 6 months regardless of gender was the best in the Spatz3 group…”
    3.    Results (Weight loss section): “Weight loss after 6 months regardless of gender was the best in the Spatz3 group…”

Note: The authors did use “highest” in some other locations, showing awareness of the issue but incomplete implementation.

Response 4:

All remaining instances of “was the best” were replaced with “was highest” or “was greatest”, as appropriate to the context.
Locations updated: (1) Abstract (sentence 3): “Weight loss after 6 months was greatest in the Spatz3 group…”; (2) Results—BMI section (para 2, sentence 1): “BMI reduction after 6 months was highest in the Spatz3 group…”; (3) Results—Weight loss section (para 1, sentence 1): “Weight loss after 6 months was highest in the Spatz3 group…”.